# STREAMINGVLM: REAL-TIME UNDERSTANDING FOR INFINITE VIDEO STREAMS

**Ruyi Xu**[1*]    **Guangxuan Xiao**[1*]    **Yukang Chen**[2]    **Liuning He**[1]

**Kelly Peng**[3]    **Yao Lu**[2]    **Song Han**[1,2]

[1]**MIT**    [2]**NVIDIA**    [3]**First Intelligence**

`https://github.com/mit-han-lab/streaming-vlm`

## ABSTRACT

Vision-language models (VLMs) could power real-time assistants and autonomous agents, but they face a critical challenge: understanding near-infinite video streams without escalating latency and memory usage. Processing entire videos with full attention leads to quadratic computational costs and poor performance on long videos. Meanwhile, simple sliding window methods are also flawed, as they either break coherence or suffer from high latency due to redundant recomputation. In this paper, we introduce **StreamingVLM**, a model designed for real-time, stable understanding of infinite visual input. Our approach is a unified framework that aligns training with streaming inference. During inference, we maintain a compact KV cache by reusing states of attention sinks, a short window of recent vision tokens, and a long window of recent text tokens. This streaming ability is instilled via a simple supervised fine-tuning (SFT) strategy that applies full attention on short, overlapped video chunks, which effectively mimics the inference-time attention pattern without training on prohibitively long contexts. For evaluation, we build **Inf-Streams-Eval**, a new benchmark with videos averaging over two hours that requires dense, per-second alignment between frames and text. On Inf-Streams-Eval, **StreamingVLM** achieves a **66.18%** win rate against GPT-4O mini and maintains stable, real-time performance at up to 8 FPS on a single NVIDIA H100. Notably, our SFT strategy also enhances general VQA abilities without any VQA-specific fine-tuning, improving performance on LongVideoBench by +4.30 and OVOBench Realtime by +5.96.

## 1 INTRODUCTION

VLMs could power autonomous driving, embodied agents, and real-time assistants, but they face critical challenges: understanding near-infinite video, responding in real time stably. To accept infinite input, common ideas are Sliding Window Attention with or without overlapping. As shown in Figure 1: (a) *Full Attention* suffers from heavy memory and latency; (b) *Sliding Window (w/o Overlapping)* resets context frequently and breaks coherence; (c) *Sliding Window Attention (w/ Overlapping)* keeps recent tokens but recomputes attention many times, which hurts efficiency.

Aligning training with inference adds further challenges. Real streaming requires taking infinite visual input in real time and replying with very low delay, but training cannot use extremely long videos. Current approaches to KV cache eviction often lack alignment with the training phase. How to train on short videos and still enable the model to reason over very long streams remains underexplored. This leads to our core question: *How can we train VLMs to understand video chunks in real time and reason stably over infinite video, moving toward human-like intelligence?*

In this paper, we propose **StreamingVLM**, a unified framework that aligns training with streaming inference and a dataset curation pipeline. The key ideas are: (1) Train the VLM with full attention on short, overlapped video chunks. (2) At inference, use an attention sink and a sliding window with to handle infinite video, aligned with training. (3) Reuse past KV states and use contiguous position IDs to keep inference stable.

---

*Equal contribution

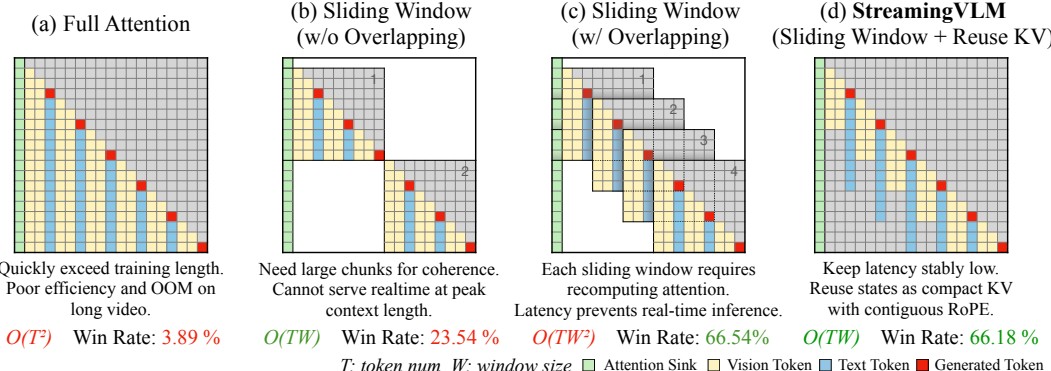

Figure 1: **Illustration of StreamingVLM vs. existing VLMs.** Let $T$ be video length and $W$ the sliding-window size. (a) *Full Attention*: $O(T^2)$ cost; unbounded memory; degrades beyond training length. (b) *Sliding Window (no overlap)*: bounded memory but short chunks break coherence; long chunks raise latency. (c) *Sliding Window (overlap)*: recomputation per window yields high latency. (d) *StreamingVLM* (Sliding Window + Reuse KV): reuses states of attention sinks, a short vision window and long text window, preserving history at low latency. "Win rate" is the pairwise win share vs. GPT-4o mini (judge: GPT-5).

Using this framework, we build **Inf-Streams-Train**, a sports commentary SFT dataset of over 4000 hours and **Inf-Streams-Eval**, a new benchmark with videos averaging over two hours that requires dense, per-second alignment between frames and text. Then, we fine-tune Qwen-2.5-VL-7B-Instruct for real-time commentary, yielding StreamingVLM that can understand infinite video and response in real time. We evaluate StreamingVLM on captioning and VQA tasks, including LiveCC-Sports-3K CC and Inf-Streams-Eval for captioning, and LongVideoBench (and related VQA benchmarks) for video understanding (Chen et al., 2025a; Wang et al., 2025a).

On captioning tasks, StreamingVLM, with its infinite video understanding, outperforms existing models such as Livecc-7B-Instruct. As shown in Figure 2, StreamingVLM performs well on practical tasks: it can provide continuous commentary for more than two hours on sports games. On VQA tasks, even without any VQA fine-tuning, StreamingVLM still improves on LongVideoBench by +4.30. In terms of efficiency, StreamingVLM maintains a low and stable latency, making it highly suitable for real-world streaming understanding tasks.

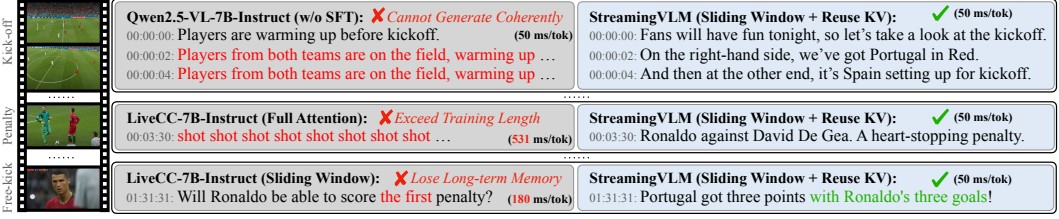

Figure 2: **Issues with existing VLMs.** (1) Without SFT, models cannot generate cross-round content coherently. (2) With full attention, the context exceeds the training length after processing 2–5 minutes of video and latency becomes prohibitive. (3) With a sliding window, models cannot retain enough context to benefit from efficiency. In contrast, StreamingVLM addresses these issues, enabling coherent commentary, real-time generation, and long-term history.

## 2 METHOD

In this section, we introduce our method for the model and the data. This part has three components: (1) inference scheme for vision–language processing that supports low-latency updates on infinite video used by **StreamingVLM**; (2) a training strategy that equips **StreamingVLM** with streaming inference capability; and (3) the data curation pipelines that provides long-horizon, real-time data for training and a new benchmark, **Inf-Streams**.

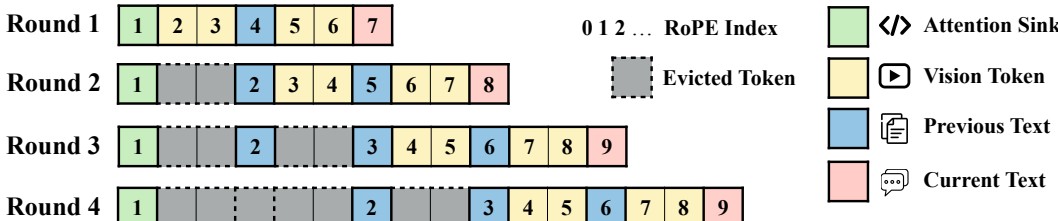

Figure 3: **Inference scheme of StreamingVLM.** We keep 512 attention-sink tokens to stabilize attention, a long text window of 512 recent tokens to preserve long-term memory, and a short vision window covering 16 seconds to track ongoing actions. We use *Contiguous RoPE*: indices are shifted to stay within a fixed range, keeping positions in-distribution and within the training length.

## 2.1 INFERENCE SCHEME OF STREAMINGVLM

This section describes the StreamingVLM inference structure shown in Figure 3. These design choices reduce the computation in Figure 1(c) while maintaining comparable performance.

**Streaming-aware KV Cache**  The key idea is to maintain a compact and stable KV cache by reusing previous states during streaming inference. As new video frames arrive, we **reuse** the states of (i) a set of sink text tokens — including the system and previous text — of length $T_{\text{sink}}$; (ii) a long window of the most recent text tokens of length $T_{\text{window}}$; and (iii) a short window of the most recent vision tokens of length $V_{\text{window}}$. In Figure 3, the cache lengths are $T_{\text{sink}} = 1$, $T_{\text{window}} = 3$, and $V_{\text{window}} = 4$.

With this structure, older vision tokens are evicted first; early text is evicted only when the budget is exceeded. Instead of recomputing previous tokens, this asymmetric retention keep the lowest computation while maintaining sufficient context for coherent generation over time, yielding comparable performance with Sliding Window with Overlapping (Figure 1(c)).

**Contiguous RoPE**  To prevent positional drift after eviction, we apply contiguous rotary positional embeddings (RoPE). When earlier tokens are removed, the RoPE indices of subsequent and incoming tokens are shifted so that their positions remain numerically contiguous with the last retained token. Once the video length surpasses the total window size, the effective RoPE indices stop growing and remain within a bounded range. This keeps positional values in-distribution and stabilizes long-horizon streaming inference.

When applied to the Qwen-VL family, which uses 3D positional embeddings for visual tokens, we use *contiguous 3D RoPE*. The RoPE index is still left-shifted to stay contiguous; for vision tokens, we build 3D indices (time, height, width) and assemble them by the 3D rule, matching the interleaved vision–text layout.

## 2.2 TRAINING STRATEGY

To endow the model with the ability to follow the streaming inference pattern in Figure 3 while keeping training simple, we adopt an *overlapped-chunk, full-attention* strategy (see Figure 4). The left panel of Figure 4 illustrates the attention at inference time. In this Figure 4, the cache lengths are the same to Figure 3, with $T_{\text{sink}}$=1, $T_{\text{window}}$=3, and $V_{\text{window}}$=4.

During training (middle panel of Figure 4), rather than replicating the exact sliding-window schedule used at inference, we split a long video stream into consecutive chunks $\{\mathcal{C}_1, \mathcal{C}_2, \ldots\}$ of length $W$ frames, with temporal overlap $O$ frames between $\mathcal{C}_i$ and $\mathcal{C}_{i+1}$ ($0 < O < W$). Each chunk is treated as a training instance in which vision and text tokens (V/T) are sampled and interleaved at 1 s intervals. We apply full attention within a chunk, i.e., every token may attend to all tokens inside the same chunk.

As highlighted in the right panel of Figure 4, this overlapped full-attention supervision closely approximates the effective attention pattern at inference — attention sink, a longer window of recent text, and a shorter window of recent vision retained in the compact KV cache. Aligning training

supervision with the test-time context teaches the model the intended recency bias and yields stable streaming behavior without training on prohibitively long, quadratic-cost contexts.

Importantly, mirroring the inference-time schedule, we interleave vision and text tokens within each training chunk — rather than adopting the common VLM paradigm that places all vision tokens before text. We compute loss only on text positions aligned to the per-second narration; when a second has no narration, we insert a placeholder token `"..."` in that slot while keeping the interleaved V/T layout. This supervision teaches the model to synchronize generation with the stream—learning when to speak and when to remain silent—and consequently endows StreamingVLM with reliable streaming narration behavior at inference.

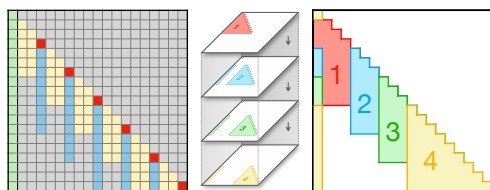

Figure 4: **Training Strategy.** We train with *overlapped full attention* that mimics test-time attention. (1), (2), (3) and (4) are four training samples, both keeping the attention sinks and overlap later in time.

## 2.3 DATA CURATION PIPELINE

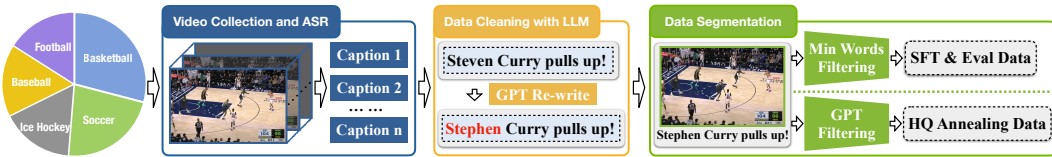

Figure 5: **Data Curation Pipeline.** We collect games from five sports—basketball, soccer, American football, ice hockey, and baseball. We use GPT to edit or reject low-quality segments, yielding 2,449 full games. We then build two datasets through separate pipelines: an SFT dataset using overlapped chunking, and a high-quality annealing dataset focused on real-time actions.

### 2.3.1 VIDEO COLLECTION AND ASR

As shown in Figure 5, we collected game videos from five sports: basketball, soccer, ice hockey, baseball, and American football, including 712 basketball games, 544 soccer games, 402 ice hockey games, 399 baseball games, and 392 American football games. The commentary language is English. To ensure video quality and read speed, we constrained the video resolution to 360P–720P with a frame rate of 24 FPS. First, we used the WhisperX model to extract real-time speech (ASR) from these games, obtaining an initial corpus of videos with a total duration of over 6,000 hours and their corresponding real-time commentary.

### 2.3.2 DATA CLEANING

In complete commentary videos, there are often many useless segments, such as advertisements and host monologues. These segments have weak connections between visual content and ASR semantics, making it impossible for the model to infer content from the footage. In addition, the ASR model sometimes fails to correctly recognize details such as player names and team names.

Therefore, we set rules and used GPT to clean these data. We first split a game into 120-second segments and concatenate the commentary within each segment, then split it into sentences. Using the segment and the video title (including game time and both teams) as context, we ask the GPT-5 model to make a decision according to the rules, with options "keep," "delete," and "edit" each sentence in one chunk. "Keep" means the content is game commentary and is correct. "Edit" means it is commentary but needs to modify some details, such as incorrect names, and the corrected complete sentence is returned. "Delete" means non-compliant content that should not appear in the training data.

For kept sentences, the timestamps are consistent with the ASR results; for edited sentences, we evenly distribute the original sentence duration over each word of the edited sentence (since a sentence typically lasts about 3–5 seconds, the error is within a tolerable range). In the original ASR data, 46.32% were kept, 37.89% were edited, and 15.79% were deleted, ultimately forming the raw video-commentary pairs of our data.

### 2.3.3 SFT AND EVALUATION DATA SEGMENTATION

**For the train and validation sets**, we build the data as follows. Under the training setup in Section 2.2, we split videos with $W = 24$ s and $O = 12$ s. To ensure enough commentary labels per sample, we require at least $2 * W$ words as min words filtering. All commentary before the segment is treated as previous text. During training, we take the first $T_{\text{sink}}$ tokens and the last $T_{\text{window}}$ tokens from this previous text to match the inference setup.

**For evaluation**, we create a new benchmark, Inf-Streams-Eval. It contains 20 full games with an average length of 2.12 hours. We split each game into 100 s segments, selecting those with at least 200 words. Commentaries of these segments are considered as ground truth. For scoring, a larger model (we use `gpt-5`) votes between two model outputs with access to ground-truth references. The model with more votes (higher win rate) is judged to provide better commentary.

Inf-Streams-Eval has two settings: *chunk* and *infinite*, denoted by $^{\dagger}$ and $^{\infty}$, respectively in following tables. In Figure 1, the chunk mode is panel (b), and the infinite mode is panel (d). For models that cannot do infinite inference, we cut the video into chunks; the model receives the previous text and the current chunk to produce a caption. For models that support infinite inference, the model runs on the full stream; we keep its past outputs as previous text and continue captioning until the video ends.

### 2.3.4 HIGH-QUALITY ANNEALING DATA

The above dataset can sft the model's ability for real-time video understanding. However, it contains a lot of content such as team information and season history; for the human experience of the commentary task, we prefer the model to provide real-time commentary on on-field events. Therefore, we created a high-quality annealing data.

We first slice all data without overlap, requiring each clip to be 16–64 seconds long with internal silence no longer than 3 seconds; each clip also contain at least $2 * D$ (duration in seconds) words. Across all games, we obtained 52,530 new samples. Then, we define the standard of "real-time commentary." For each sample, we use GPT-5 to determine whether the proportion of "real-time commentary" exceeds 80% to decide whether to keep it. In the end, only 14,786 samples were retained. Subsequent experiments in Table 6 show that after applying this portion of data for sft, the model's capability and commentary quality further improved.

## 3 EXPERIMENTS

In this section, we first describe the implementation details, then evaluate on video captioning and VQA against strong baselines. We next test the efficiency of StreamingVLM. Finally, we run ablations to better understand its behavior.

### 3.1 EXPERIMENTAL SETUP

**Training** We fine-tune StreamingVLM from Qwen2.5-VL-Instruct-7B (Bai et al., 2025). Step 1 teaches the model the infinite streaming inference pattern. We train on our SFT set (525K streaming samples) and on LiveCC's Live-WhisperX-526K (526K streaming samples) (Chen et al., 2025a). Step 2 uses our high-quality annealing data (14K streaming samples, each 16–64 s with detailed actions) to boost real-time action commentary and improve human experience. After these two stages, we obtain StreamingVLM. The total compute is about 128 H100-days.

**Baselines** We select strong baselines to compare with StreamingVLM. For the captioning task, we use GPT-4o mini to show commentary strength, and Livecc-7B-Instruct, which is trained on 5.5M YouTube video clips ($30 - 240$ s) and 178K Video-Question-Answer samples, working well on short videos commentary (OpenAI, 2024; Chen et al., 2025a). We also include ReKV, a strong training-free streaming-inference method (Di et al., 2025). Due to design limits, GPT-4o mini is evaluated on Inf-Streams-Eval in the *chunk* setting, not the infinite mode used by StreamingVLM. LiveCC-7B-Instruct is tested in both *chunked* and *infinite* settings. For the VQA task, we use Qwen2.5-VL-7B-Instruct, which is the base model before SFT for StreamingVLM, to show that our SFT pipeline improves the base ability (Bai et al., 2025).

Table 1: **Captioning accuracy** (win rate vs. baselines). Baselines with/without chunking fall short; StreamingVLM surpasses strong models such as GPT-4o and produces compelling commentary.(Superscripts for Inf-Streams-Eval: $\infty$ = infinite; $\dagger$ = chunk length 100s. On Livecc-Sports-3K CC, LiveCC has only one mode and cannot be compared against itself, so we show "–".

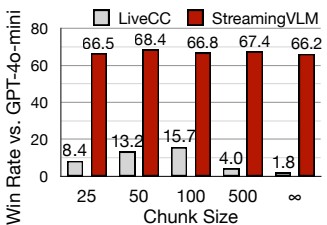

Figure 6: For existing VLMs, balancing cross-chunk coherence with training-length limits is challenging.

| Win Rate A vs. B | Inf-Streams-Eval | | | Livecc-Sports-3K cc | | | |
|---|---|---|---|---|---|---|---|
| Model B ⟍ Model A | GPT-4o$^\dagger$ | Livecc$^\dagger$ | Livecc$^\infty$ | LLaVA | GPT-4o | Gemini | Livecc |
| Qwen-2.5-VL-7B-Instruct $^\dagger$ | 0.01 | 20.44 | 95.97 | 24.50 | 16.25 | 28.38 | 34.11 |
| Livecc-7B-Instruct $^\dagger$ | 15.73 | – | – | – | – | – | – |
| Livecc-7B-Instruct $^\infty$ | 1.82 | – | – | 41.50 | 40.06 | 39.73 | – |
| StreamingVLM $^\infty$ | **66.18** | **87.81** | **99.12** | **47.33** | **45.59** | **44.21** | **56.19** |

**Benchmark** We evaluate real-time captioning and video understanding across a broad set of tasks. For captioning, we use our Inf-Streams-Eval (average length 2.12 hours), which tests long-horizon commentary and the LiveSports3K-CC benchmark (49 sports, 416 clips, each $\geq$ 10 s) (Chen et al., 2025a). For video understanding, we evaluate StreamingVLM on four public suites. VideoMME: a multi-task set (QA, caption, grounding) covering short and long videos for general comprehension (Fu et al., 2025). MVBench: fine-grained skills on short clips (actions, objects, counting, temporal order) (Li et al., 2024b). LongVideoBench: long-video QA that requires long-term memory and cross-segment reasoning (Wang et al., 2025a). OVOBench: video QA that tests real-time understanding and streaming perception (Li et al., 2025).

## 3.2 ACCURACY RESULTS

### 3.2.1 CAPTIONING

We first compare our inference strategy with ReKV on the captioning task. We observe a paradox for training-free ReKV: models without task-specific fine-tuning perform poorly, yet models that are specially fine-tuned (e.g., StreamingVLM) rely on a fixed context format that ReKV's eviction policy disrupts, often yielding no output. In contrast, StreamingVLM 's training–inference consistent design resolves this issue.

Then, we evaluate StreamingVLM, Qwen-2.5-VL-7B-Instruct, and LiveCC-7B-Instruct on LiveCC-3K-Sports-CC and Inf-Streams-Eval. As shown in Table 1, on Inf-Streams-Eval, Qwen-2.5-VL-7B-Instruct cannot keep continuous commentary and thus performs poorly. LiveCC-7B-Instruct works

Table 2: **Training–inference consistency surpasses ReKV.** Non–fine-tuned models lack capability of real-time captioning, while with fine-tuning models ReKV's eviction policy disrupts context, frequently resulting in no output. (Superscripts for Inf-Streams-Eval: $\infty$ = infinite; $\dagger$ = chunk length 100s.)

| Win Rate | Inf-Streams-Eval | | |
|---|---|---|---|
| Model B ⟍ Model A | GPT-4o$^\dagger$ | Livecc$^\dagger$ | Livecc$^\infty$ |
| Qwen (+ ReKV) $^\infty$ | 0.00 | 19.56 | 63.57 |
| StreamingVLM (+ ReKV) $^\infty$ | 0.00 | 0.00 | 0.00 |
| StreamingVLM (+ Ours) $^\infty$ | **66.18** | **87.81** | **99.12** |

better with *chunked* inference. Figure 6 further shows that short chunks break coherence; these designs do not support infinite inference, and with long chunks they soon exceed the training length and degrade.

In contrast, StreamingVLM runs in infinite mode; its long-term memory and streaming video perception give it a clear edge, surpassing GPT-4o mini in commentary quality. Figure 2 (the figure shown) illustrates a real case where StreamingVLM maintains coherent output, real-time latency, and long-term memory, addressing the core challenge of real-time perception for infinite video streams. On LiveCC-3K-Sports-CC, StreamingVLM also performs better than baselines, showing stable streaming captioning on videos of various length.

### 3.2.2 VQA

We evaluate StreamingVLM and its base model, Qwen-2.5-VL-7B-Instruct, on four VQA tasks. As shown in Table 3, even without any VQA SFT, StreamingVLM outperforms the base on all tasks,

Table 3: VQA results comparing StreamingVLM with its base model. Without any VQA fine-tuning, StreamingVLM delivers consistent accuracy gains across all tasks, with the strongest improvements on long-horizon and real-time settings.

|  | MVBench | Video MME (w/o sub.) | LongVideoBench | OVOBench (Realtime) |
|---|---|---|---|---|
| Qwen-2.5-VL-7B-Instruct | 67.34 | 65.10 | 54.70 | 56.00 |
| StreamingVLM | **69.16** | **65.10** | **59.00** | **61.96** |

Table 4: Ablation of RoPE on captioning (win rate). Native RoPE drops on infinite streams; 100 s chunking partly recovers but hurts long-term memory; contiguous RoPE keeps indices bounded and sustains infinite performance. (Superscripts for Inf-Streams-Eval: $\infty$ = infinite; $\dagger$ = chunk length 100s.)

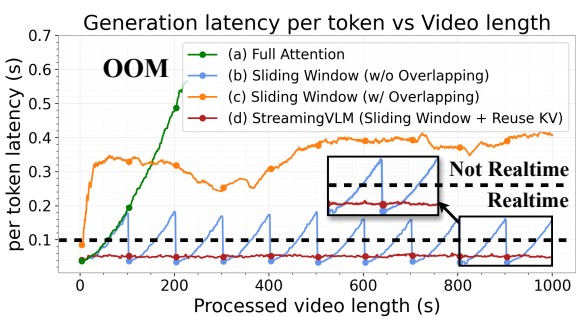

| Win Rate A vs. B | Inf-Streams-Eval | | |
|---|---|---|---|
| Model B / Model A | GPT-4o$^\dagger$ | Livecc$^\dagger$ | Livecc$^\infty$ |
| Native $^\dagger$ | 63.23 | 74.00 | 98.07 |
| Native $^\infty$ | 25.09 | 59.42 | 60.32 |
| Contiguous $^\infty$ | **66.18** | **87.81** | **99.12** |

Figure 7: Per-token latency vs. video length. Full attention hits OOM; sliding window w/o Overlapping spikes above real time; sliding window w/ Overlapping remains inefficient; StreamingVLM latency stays low and stable. The dashed line marks the real-time threshold (10 tokens/s $\Rightarrow \leq 0.1$ s per token).

showing that our SFT improves general visual ability. OVOBench Realtime tests understanding of the immediate, streaming scene. On this streaming perception task, StreamingVLM improves by **5.96%**. This highlights the strength of Inf-Streams-Train and our training strategy, which enhances the model's core abilities.

### 3.3 EFFICIENCY TESTS

As shown in Figure 7, we report per-token latency for the three methods in Figure 1 on infinite commentary: VLMs with full attention, sliding window attention (w/o overlapping), sliding window attention (w/ overlapping), and the inference strategy of StreamingVLM, respectively correspond to panels (a), (b), (c), and (d) in the Figure 1.

Real-time replies require latency below a fixed threshold as the dashed line. Full attention soon exceed the limit and OOM. Sliding window (w/o overlapping) needs large chunks for coherence, so it shows a periodic latency pattern: at the start of each chunk the model rebuilds context and the commentary is not coherent with the past; later in the chunk, latency rises sharply and fails to meet real-time needs. Sliding window (w/ overlapping) remains inefficient for computation redundancy. StreamingVLM keeps fixed context length and reuses KV, maintains lower and stable latency, and supports real-time commentary at 8 FPS on a single NVIDIA H100.

### 3.4 ABLATION STUDY

#### 3.4.1 CONTIGUOUS ROPE

We study the effect of contiguous RoPE indices. Since we train with full attention, training only uses the native RoPE. At inference, we compare contiguous RoPE with the native version. As shown in Table 4, native RoPE degrades sharply on infinite streams because its index grows fast and exceeds the training range. Splitting the video into 100 s chunks can partly recover accuracy, but it harms long-term conherence. With *contiguous RoPE*, the position index stays bounded, so the model supports infinite inference without loss.

Table 5: Ablation of sliding window and sink size with accuracy on captioning tasks (win rate). **Left**: effect of $T_{\text{sink}}$ and $T_{\text{window}}$, trained with $V_{\text{window}} = 16\,\text{s}$. **Right**: effect of $V_{\text{window}}$, trained with $T_{\text{sink}} = 512$ and $T_{\text{window}} = 512$. (Superscripts for Inf-Streams-Eval: $\infty$ = infinite; $\dagger$ = chunk length 100s. )

| Infer args | | SFT args | | Inf-Streams-Eval (Basketball) | | |
|---|---|---|---|---|---|---|
| $T_{\text{sink}}$ | $T_{\text{window}}$ | $T_{\text{sink}}$ | $T_{\text{window}}$ | GPT-4o$^\dagger$ | Livecc$^\dagger$ | Livecc$^\infty$ |
| 512 | 0 | 512 | 512 | 69.68 | 89.42 | 99.19 |
| 0 | 512 | 512 | 512 | 66.76 | 86.03 | 98.69 |
| 256 | 256 | 512 | 512 | 70.17 | 91.79 | 99.62 |
| 1024 | 1024 | 512 | 512 | 71.43 | 91.69 | **99.84** |
| $\infty$ | $\infty$ | $\infty$ | $\infty$ | 60.41 | 72.08 | 98.55 |
| 512 | 512 | 512 | 512 | **73.64** | **92.33** | 99.38 |

| $V_{\text{window}}$ | Inf-Streams-Eval | | |
|---|---|---|---|
| Win Rate vs. | GPT-4o$^\dagger$ | Livecc$^\dagger$ | Livecc$^\infty$ |
| 0 s | 52.90 | 77.49 | 97.56 |
| 1 s | 63.46 | 83.24 | 98.18 |
| 4 s | 66.08 | 83.86 | 98.73 |
| 8 s | 65.66 | 85.09 | 99.14 |
| 32 s | 65.49 | 85.58 | 99.06 |
| 16 s | **66.18** | **87.81** | **99.38** |

Table 6: Ablation of SFT strategy and dataset on captioning and VQA. Overlapped SFT strategy improves over the Live-WhisperX-526K base, and adding the high-quality annealing data brings further improvements, especially for infinite streaming task Inf-Streams-Eval. (Superscripts for Inf-Streams-Eval: $\infty$ = infinite; $\dagger$ = chunk length 100s.)

| Win Rate A vs. B Model B / Model A | Inf-Streams-Eval | | | Livecc-Sports-3K cc | | | | MVBench | Video MME w/o sub. | LongVideoBench | OVOBench Realtime$^\dagger$ |
|---|---|---|---|---|---|---|---|---|---|---|---|
| | GPT-4o$^\dagger$ | Livecc$^\dagger$ | Livecc$^\infty$ | LLaVA | GPT-4o | Gemini | Livecc | Score | | | |
| Qwen-2.5-VL-7B-Instruct $^\dagger$ | 0.01 | 20.44 | 95.97 | 24.50 | 16.25 | 28.38 | 34.11 | 67.34 | **65.10** | 54.70 | 56.00 |
| + Live-WhisperX-526K $^\infty$ | 32.17 | 56.52 | 99.05 | 42.77 | 41.86 | 39.37 | 47.80 | 63.71 | 62.10 | 54.30 | 57.69 |
| + Inf-Streams-Train $^\infty$ | 63.46 | 83.82 | 98.95 | 46.45 | 45.48 | 44.27 | 53.07 | 68.66 | 64.90 | **59.00** | 60.55 |
| + High-Quality Annealing Data $^\infty$ | **66.18** | **87.81** | **99.12** | 47.33 | 45.59 | 44.39 | 56.19 | 69.16 | 65.10 | 59.00 | 61.96 |

### 3.4.2 SLIDING WINDOW AND SINK

We firstly verify the value of evicting text during training. Then we search for the best inference settings of $T_{\text{sink}}, T_{\text{window}}, V_{\text{window}}$.

First, the left table in Table 5 ablates the lengths of the attention sink and text window. Here $T_{\text{sink}}$ and $T_{\text{window}}$ are the lengths of previous attention sink and text window kept during both training and inference. We take a basketball-only subset of the SFT data and train two models: one with text eviction using $T_{\text{sink}}=512$ and $T_{\text{window}}=512$, and one without eviction. On the Inf-Streams-Eval (basketball subset), we evaluate each model under its matching policy (evict vs. no-evict). The left table in table 5 shows that, for infinite inference, evicting previous text tokens is important and improves performance.

Next, we study different choices of $V_{\text{window}}$. The right table in Table 5 shows that a 16 s visual window is a good choice: it is long enough to cover recent actions, yet short enough to stay efficient. In contrast, keeping 0 s of vision context leads to a clear drop, confirming that retaining recent vision tokens for continuous actions is essential.

### 3.4.3 TRAINING STRATEGY AND DATASET

We study the effect of our SFT data and high-quality annealing data. The SFT set teaches the model the infinite streaming inference pattern, while the high-quality annealing data further improves commentary quality.

**SFT Strategy** As shown in Table 6, with our overlapped training strategy, our SFT subset helps the model adapt to the interleaved vision–text pattern and to understand very long videos. Compared with a model trained only on Live-WhisperX-526K, training on the overlapped SFT data strengthens perception of infinite video, yielding clear gains +31.29 (win rate against GPT-4o-mini) on Inf-Streams-Eval and +3.68 (win rate against LLaVA-Video-72B-Qwen2) on Livecc-Sports-3K cc.

As shown in Table 7, we train a new baseline model using our full SFT dataset but without our overlapped-chunk SFT strategy (i.e., using standard non-overlapping chunks). We hypothesize two things: (1) This new model performs significantly worse, proving the data alone is insufficient. (2)

Table 7: Ablation of SFT strategy on captioning and VQA. Overlapped SFT strategy performs significantly better than the ablation model trained without our overlapped-chunk SFT strategy. (Superscripts for Inf-Streams-Eval: $\infty$ = infinite; $\dagger$ = chunk length 100s.)

| Win Rate A vs. B | Inf-Streams-Eval | Livecc-Sports-3K cc | | | | MVBench | Video MME | LongVideoBench | OVOBench |
|---|---|---|---|---|---|---|---|---|---|
| Model B ⟍ Model A | GPT-4o$^\dagger$ | LLaVA | GPT-4o | Gemini | Livecc Score | | w/o sub. | | Realtime |
| Non-overlapping Strategy $^\infty$ | 62.51 | 46.24 | 45.08 | 43.21 | **56.19** | 68.79 | **65.50** | 58.90 | 59.20 |
| Overlapped Strategy $^\infty$ | **66.18** | **47.33** | **45.59** | **44.39** | **56.19** | **69.16** | 65.10 | **59.00** | **61.96** |

This model will fail or perform poorly when run with our streaming KV cache, proving our SFT strategy is essential for the training-inference alignment.

**High-quality Annealing Data** Our high-quality annealing data focus on real-time content and further boosts model ability. As shown in Table 6, we compare training with and without the high-quality annealing data. We can observe significant gains on both captioning and VQA benchmarks.

# 4 RELATED WORK

**Vision–Language Models** Early multimodal models start from images and then extend to videos by adding temporal modules or token schedulers. Recent open models improve video understanding and transfer across tasks. Examples include LLaVA-OneVision for unified transfer across images, multi-image inputs, and videos (Li et al., 2024a), Video-LLaMA 2 for spatial–temporal and audio cues (Cheng et al., 2024), InternVideo2/2.5 for scaling video encoders and long context (Wang et al., 2024; 2025b), LongVILA for long video training system (Chen et al., 2025b), and Qwen2.5-VL for strong grounding, document parsing, and long-video skills (Bai et al., 2025). Most systems process finite clips and often place all vision tokens before text, which can hurt alignment in streaming and limit real-time interaction in practice. In contrast, we interleave vision and text at 1 s steps to match real-time commentary and interaction, and we observe gains on both commentary and VQA.

**Long-Context and Streaming Inference in Text LLMs** To handle near-infinite inputs under fixed memory and delay, the text community has proposed several lines of work: (1) *Attention sink + sliding window:* StreamingLLM keeps a small set of early "sink" tokens plus a recent window, which stabilizes very long decoding (Xiao et al., 2024). (2) *RoPE extension and continuity:* YaRN, LongRoPE, and LongLoRA for efficient fine-tuning improve position embedding extrapolation (Peng et al., 2023; Ding et al., 2024; Chen et al., 2024b); our contiguous RoPE follows this idea but targets cross-modal, step-wise updates. (3) *KV cache compression/eviction:* $H_2O$, SnapKV, and ReKV reduce KV size by selecting heavy hitters or gating heads (Zhang et al., 2023; Li et al., 2024c; Di et al., 2025). However, these methods are mostly tested on text, and alignment between streaming training and inference remains underexplored. We bring the "sink + sliding window + contiguous position" recipe to cross-modal streaming and introduce a training strategy for streaming inference.

**Streaming and Online Video LLMs** Several concurrent works target streaming video directly. VideoLLM-online (LIVE) converts offline data into streaming dialogue for long context and low latency (Chen et al., 2024a). VideoStreaming uses a fixed video token budget to handle long videos (Qian et al., 2024). LiveCC aligns large-scale ASR with video frames to push real-time sports commentary (Chen et al., 2025a). In practice, on videos longer than 5 minutes (at least 200 frames), these methods show clear performance drops, and their latency is still far from infinite real-time interaction. Compared with these, we (i) train with *overlapped short chunks and full attention* to match the *sink + sliding window* test pattern, and (ii) keep *contiguous RoPE* across modalities to enable real-time understanding over infinite videos.

**VLMs Benchmarks and Evaluation** VideoMME covers 900 videos (254 hours) with multimodal inputs and tests both short and long time ranges (Fu et al., 2025). LiveSports-3K-CC compares real-time commentary quality and often uses the "LLM-as-a-judge" win-rate metric (Wang et al., 2025a). LVBench targets ultra-long videos and long-term memory (Wang et al., 2025a). However, Current benchmarks often focus on retrieval or summary over long videos and do not require frame-level understanding, so even a very low FPS sample may pass. Our Inf-Streams-Eval is built for near-*infinite*

commentary (over 2 hours). It requires second-level alignment between frames and responses and tests high-FPS, long-video understanding—closer to real-world needs for VLM assistants, robots, and autonomous driving.

## 5 CONCLUSION

In this paper, we introduce StreamingVLM, a unified training–inference framework that brings real-time streaming perception to existing VLMs. We first present an efficient strategy for training streaming VLMs and a data curation pipeline that together boost performance on both streaming tasks and VQA. We then show on real-world cases that our inference design enables real-time video understanding, delivering stable commentary for over 3 hours at up to 8 FPS on a single NVIDIA H100. Finally, we release Inf-Streams, a new SFT dataset and benchmark that tests second-level, real-time understanding on videos averaging over 2 hours. Taken together, this work paves the way for practical deployment in real settings.

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

## A APPENDIX

### A.1 LLM USAGE STATEMENT

We acknowledge the use of Large Language Models (specifically Claude and GPT-5) in the preparation of this manuscript. The LLMs were used exclusively as writing assistants to:

- Polish and refine the language for clarity and conciseness
- Improve grammar and sentence structure
- Suggest alternative phrasings for technical descriptions
- Help organize and structure sections for better flow

All research ideas, experimental design, theoretical derivations, and scientific contributions are entirely our own. The LLMs did not contribute to research ideation, hypothesis formulation, or any core scientific aspects of this work. We used LLMs in a manner similar to grammar-checking tools, but with more sophisticated language capabilities. All content, including any LLM-assisted text, has been carefully reviewed and verified by the authors. We take full responsibility for all contents of this paper, including their accuracy and originality.

### A.2 STABILITY OVER TIME

We split each video into five segments at 20% intervals and evaluate on the 2-hour test set. As shown in Figure 8, StreamingVLM does not degrade across later segments and reaches performance close to Sliding-Window w/ Overlap. This indicates that StreamingVLM maintains quality as videos grow and effectively supports unbounded inference.

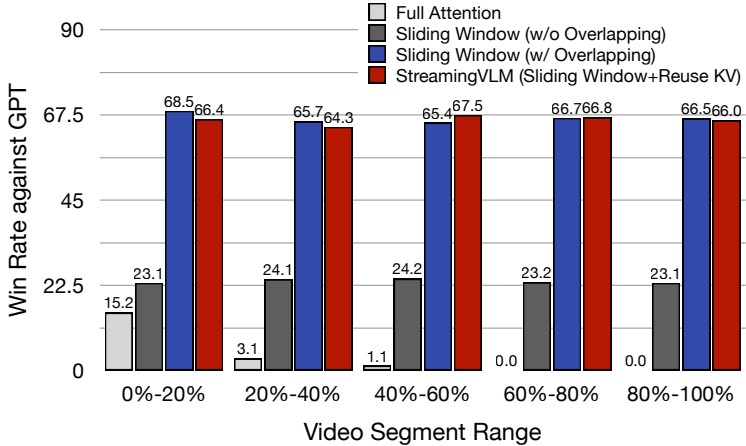

Figure 8: **Stability over time.** Each test video is split into five segments at 20% intervals. StreamingVLM (Sliding Window + Reuse KV) maintains nearly constant win rate across segments and matches the performance of Sliding Window w/ Overlap, while Full Attention and Sliding Window w/o Overlap degrade or remain far lower.

Table 8: Sensitivity analysis over $Tsink$ on Inf-Stream-Eval.

| SFT $T_{\text{sink}}$ | Eval $T_{\text{sink}}$ | GPT-4o[†] | Livecc[†] | Livecc[∞] |
|---|---|---|---|---|
| 64 | 64 | 72.04 | 90.32 | 99.46 |
| 128 | 128 | 73.65 | 92.94 | 99.47 |
| 256 | 256 | 73.73 | 93.28 | 99.38 |
| 1024 | 1024 | 74.82 | 93.51 | 99.53 |
| 512 | 512 | 73.64 | 92.33 | 99.38 |

### A.3 Sensitivity Analysis of Sink Token Window Size

To address how finite token length limits impact model performance across varying scenarios, we conducted a sensitivity analysis on the attention-sink window size ($T_{\text{sink}}$). As discussed in the Limitations section, these window sizes serve as key hyperparameters.

Table 8 presents the ablation results for varying $T_{\text{sink}}$ sizes (64, 128, 256, 512, and 1024) during both Supervised Fine-Tuning (SFT) and Evaluation. The results demonstrate that the sink token size noticeably impacts the final performance. Generally, larger $T_{\text{sink}}$ capacities yield better win rates against GPT-4o and Livecc metrics, as the model retains more initial contextual tokens.

However, performance gains plateau at larger window sizes, indicating a trade-off between context retention and computational efficiency. This confirms that $T_{\text{sink}}$ should be carefully tuned based on the specific scenario's context length requirements.

### A.4 Demo

We provide a demo video in the supplementary materials showing StreamingVLM 's commentary after 100 minutes of continuous inference. The video is randomly selected and edited to remove long pauses and mid-length ads. As the base model is modest in size, occasional hallucinations may occur. Please see the supplementary materials for details.

