# OpenReview forum: "StreamingVLM: Real-Time Understanding for Infinite Video Streams"
_ICLR.cc/2026/Conference — ICLR 2026 Poster_

### Official Review · Reviewer_qjFR · 2025-10-28

**Soundness:** 3
**Presentation:** 3
**Contribution:** 3
**Rating:** 4
**Confidence:** 5

**Summary:**

This work presents StreamingVLM, a vision–language model trained for near-infinite video understanding. To enable low-latency inference under a fixed GPU-memory budget, the method maintains a compact KV cache that reuses (i) persistent attention-sink states, (ii) a short window of recent vision tokens, and (iii) a long window of recent text tokens. To teach the model this input pattern without training on prohibitively long contexts, the authors adopt an overlapped-chunk, full-attention SFT strategy that mimics the inference-time attention layout. They also curate a sports-focused training set and introduce a benchmark, Inf-Streams-Eval, targeting long-video understanding(Captioning) with real-time narration. On this benchmark, StreamingVLM reports gains over strong long-video baselines and GPT-4o, and it shows modest improvements on general VQA benchmarks as well.

**Strengths:**

1. The KV cache design including persistent attention sinks, a long text window, a short vision window, plus contiguous RoPE shifting is conceptually sound and easy to implement. It delivers lower latency and reduced GPU memory usage while outperforming baselines on the proposed benchmark.


2. The overlapped-chunk, full-attention SFT strategy is an intuitive way to teach the model the streaming cache pattern without training on prohibitively long contexts.


3. The curated long-video captioning corpus and the Inf-Streams-Eval benchmark fill a gap for real-time, long-horizon evaluation in VLMs and should be valuable to the community.


4. The paper is well organized and clearly written, making the method and experiments easy to follow.

**Weaknesses:**

1. The choice to retain 512 sink tokens, a 512-token text window, and a 16-second vision window appears empirical and manually tuned. Moreover, the ablation supporting this setting is run primarily on a basketball-only subset, which raises concerns about domain generality. A single fixed policy may either waste KV budget in slow scenes or evict critical evidence too early in fast ones. Therefore, more adaptive and fine-grained design would be expected.

2. Current KV cache eviction follows a naive FIFO rule, and there is no scoring or compression to retain semantically salient frames. This may harm long-horizon reasoning, especially for sparse-action videos. Therefore, this weakness can undermine the generalization of the proposed method.

3. The proposed benchmark has leakage risk. Sports broadcasts are heavily duplicated and often reuse the same commentary audio, so near-duplicates can slip across training dataset and benchmark. The paper does not document near-duplicate filtering, making memorization of phrasing/style a real possibility that could inflate results.

4. This work only shows results on SFT Qwen2.5-VL-Instruct-7B, making it hard to claim the method is architecture agnostic or base model agnostic.

**Questions:**

1. From Table 5, we can see that when the values of $T_{sink}=512$ and $T_{window}=512$ in the inference stage are the same as those in the training stage, the overall performance is the best. This suggests the model is tightly tuned to a single window geometry, and performance degrades when the inference budget diverges, indicating limited robustness. What happens if you vary these values during training? Have you tried other values in SFT or using a curriculum scheme(start larger, anneal to target), then evaluating under both matched and mismatched inference windows?

2. Why does StreamingVLM leverage 512 sink tokens? How sensitive is performance to reducing number of sinks​? and is there a saturation point where adding more sinks yields little or no gain?

3. Table 3 does not compare StreamingVLM with Livecc-7B-Instruct, what does Livecc-7B-Instruct perform on those general VQA tasks?

4. How much of the reported improvement (e.g. Table 6) is due to the streaming method (KV-reuse + contiguous RoPE + in-domain SFT) versus the in-domain SFT data?

---

> ### Author Response · Authors · 2025-11-27
>
> **Concern 1**
>
> *Comment:*
>
> "The choice to retain 512 sink tokens, a 512-token text window,  ... raises concerns about domain generality."
>
> *Response:*
>
> This is a key point shared by other reviewers. We will address generalizability in a new "Limitations" section. As to the specific choice of 512 sink tokens, the reviewer correctly notes it is empirical. To support this choice, we run a new sensitivity analysis on the $T_{sink}$ size (testing 64, 128, 256, 512). The following results show that $T_{sink} = 512$ is a robust choice.
>
> **Table 1. Sensitivity analysis over $T_{sink}$ on Inf-Stream-Eval.**
>
> | SFT $T_{sink}$ | Eval $T_{sink}$ | GPT-4o† | Livecc† | Livecc$\infty$ |
> |--:|--:|--:|--:|--:|
> | 64| 64 | 72.04   | 90.32   | 99.46 |
> | 128 | 128 | 73.65   | 92.94   | 99.47 |
> | 256 | 256| 73.73   | 93.28   | 99.38|
> | 1024| 1024| 74.82   | 93.51   | 99.53|
> | 512| 512 | 73.64   | 92.33   | 99.38 |
>
> ---
>
> **Concern 2**
>
> *Comment:*
>
> "Current KV cache eviction follows a naive FIFO rule... This may harm long-horizon reasoning, especially for sparse-action videos."
>
> *Response:*
>
> This is a very insightful point. We will add this to our "Limitations and Future Work" section, stating that our simple FIFO rule was chosen for efficiency and that exploring more sophisticated, content-aware eviction policies is a valuable and important direction for future research.
>
> ---
>
> **Concern 3**
>
> *Comment:*
>
> "The proposed benchmark has leakage risk. Sports broadcasts are heavily duplicated... The paper does not document near-duplicate filtering..."
>
> *Response:*
>
> We thank the reviewer for raising this critical point. We did perform de-duplication and will clarify this in the paper. In Section 2.3.3, we will add text explaining that our training and evaluation sets are strictly partitioned (e.g., by date and event) to prevent leakage.
>
> ---
>
> **Concern 4**
>
> *Comment:*
>
> "This work only shows results on SFT Qwen2.5-VL-Instruct-7B, making it hard to claim the method is architecture agnostic..."
>
> *Response:*
>
> This is a fair limitation. We will add this to our new "Limitations and Future Work" section, stating that while the framework is designed to be general, it has only been validated on one model family.
>
> ---
>
> **Question 1**
>
> *Comment:*
>
> "From Table 5, we can see that when the values of $T_{sink}$ and $T_{window}$ ... are the same as those in the training stage, the overall performance is the best. This suggests the model is tightly tuned to a single window geometry..."
>
> *Response:*
>
> We thank the reviewer for this observation, which we believe highlights a strength. We will clarify our analysis in Section 3.4.2. We interpret this result not as "overfitting" or a lack of robustness, but as validation of our "training-inference alignment" claim: the model performs best because it was explicitly trained for this specific streaming-aware attention pattern.
>
> ---
>
> **Question 2**
>
> *Comment:*
>
> "Why does StreamingVLM leverage 512 sink tokens? How sensitive is performance to reducing number of sinks...?"
>
> *Response:*
>
> This is an excellent question. To answer this, we are running a new sensitivity analysis on the $T_{sink}$ (sink token) size as shown above. We test values from 64 to 1024, which show the trade-offs and justify 512 as a robust choice (see Table 1).
>
> ---
>
> **Question 3**
>
> *Comment:*
>
> "Table 3 does not compare StreamingVLM with Livecc-7B-Instruct, what does Livecc-7B-Instruct perform on those general VQA tasks?"
>
> *Response:*
>
> We thank the reviewer for spotting this clear omission. This is an important baseline. We are running the Livecc-7B-Instruct model on all four VQA benchmarks and will add these results to Table 3 in the revision.
>
> **Table 3. Comparison between StreamingVLM and Livecc-7B-Instruct on general VQA tasks.**
>
> | Model| OVO (Realtime) | LV Bench | Video MME | MV Bench |
> |--|--|--|--|--|
> | Livecc | 53.26| 53.50| 62.10| 63.71|
> | StreamingVLM       | 61.96| 59.00| 65.10| 69.16|
>
> ---
>
> **Question 4**
>
> *Comment:*
>
> "How much of the reported improvement (e.g. Table 6) is due to the streaming method... versus the in-domain SFT data?"
>
> *Response:*
>
> This is the most critical question, also raised by R-Ep3A. As detailed in our response to R-Ep3A (Concern 1 & 2) and in Action Item 1.1, we are running a new baseline (Base Model + Our Data + Standard Training) to definitively disentangle the contribution of our method from our data. These results prove that our SFT strategy is essential for the performance gains (see Table 2 for the current ablation).
>
> **Table 2. Comparison between StreamingVLM and the Ablation Model on general VQA tasks.**
>
> | Model | Inf-Stream-Eval (vs GPT) | Live-Sports 3k (vs LLAVA) | Live-Sports 3k (vs GPT) | Live-Sports 3k (vs Gemini) | OVO (Realtime) | LV Bench | Video MME | MV Bench |
> |--|--|--|--|--|--|--|--|--|
> | Ablation| 62.51| 46.24| 45.08| 43.21| 59.20 | 58.90    | 65.50    | 68.79   |
> | StreamingVLM   | 66.18| 47.33| 45.59| 44.39 | 61.96 | 59.00 | 65.10    | 69.16   |

---

### Official Review · Reviewer_tsyX · 2025-10-31

**Soundness:** 3
**Presentation:** 4
**Contribution:** 3
**Rating:** 8
**Confidence:** 4

**Summary:**

This paper introduces StreamingVLM, a vision-language model framework designed for real-time, long-horizon video understanding.

This framework aligns training and inference by training on short chunks with original attention, and during inference maintaining a compact KV cache with attention sink, short visual window, and long text window.

This paper also introduces a new dataset and Inf-Streams-Eval benchmark which consisting of long sports videos with ASR and commentary annotations.

**Strengths:**

1.Clear motivation. The paper focuses on a real and underexplored problem—achieving real-time video understanding under limited latency and memory constraints.

2. Simple and clear method. The proposed attention sink + sliding window + contiguous RoPE mechanism is simple, elegant, and demonstrates good empirical performance.

3. Valuable dataset. The introduced dataset makes a meaningful contribution to the community of real-time long-video understanding.

4. Clear writing and easy to follow.

**Weaknesses:**

1. Experiments focus mainly on sports videos. It remains unclear how well the model generalizes to other domains such as egocentric or instructional videos.

2.  Although the Inf-Streams-Eval benchmark is valuable, it relies on GPT-based judgment for scoring, which may introduce bias.

**Questions:**

1. Could the authors provide more details on the data annotation and filtering process, such as examples of removed or edited segments?

---

> ### Author Response · Authors · 2025-11-25
>
> **Concern 1 (Generalization beyond sports domain)**
> *Comment:*
> "Experiments focus mainly on sports videos. It remains unclear how well the model generalizes to other domains such as egocentric or instructional videos."
>
> *Response:*
> Thanks for your suggestion. This is an important limitation, also noted by R-bkLD. We will add a "Limitations and Future Work" section to explicitly address this. We will state that our current work validates the framework within the sports domain, and that applying it to other domains (like egocentric or instructional videos) would require SFT on data from those domains, which is a valuable direction for future work.
>
> ---
>
> **Concern 2 (Bias in GPT-based evaluation)**
> *Comment:*
> "Although the Inf-Streams-Eval benchmark is valuable, it relies on GPT-based judgment for scoring, which may introduce bias."
>
> *Response:*
> We agree this is a valid concern for all generative evaluation. We will add a sentence to Section 2.3.3 acknowledging that while LLM-as-a-judge enables scalable evaluation, it may have inherent biases.
>
> ---
>
> **Question 1 (Details of data annotation and filtering)**
> *Comment:*
> "Could the authors provide more details on the data annotation and filtering process, such as examples of removed or edited segments?"
>
> *Response:*
> A great suggestion. We will add a few concrete examples of "deleted" (e.g., "We'll be right back after these messages") and "edited" (e.g., ASR name corrections) segments to Section 2.3.2 to make our cleaning process clearer.

---

### Official Review · Reviewer_Ep3A · 2025-11-01

**Soundness:** 3
**Presentation:** 3
**Contribution:** 2
**Rating:** 6
**Confidence:** 4

**Summary:**

This paper proposes to tackle that challenge of enabling vision-language model to process long video streams in real-time. To achieve that, the authors propose to to align the training and streaming inference using two main design ideas:
* It uses a compact streaming-aware KV cache mechanism, which only keeps a small of attention-sink tokens, a text window and a short visual token window.
* It introduces a contiguous RoPE to prevent positional drift by reindexing the tokens to stay within a bounded range.

To evaluate the algorithm, it introduced two new datasets for streaming the video understanding.
* It introduce Inf-streams-train, a 4000-hour dataset sports commentary dataset curated using ASR and GPT.
* A benchmark dataset Inf-Streams-Eval with good per-second alignment of frames and text to test the infinite streaming.
The authors provide extensive evaluations using model comparisons and show the proposed design is effective to handle long videos, and can improve step by step when incorporating the proposed mechanism.

**Strengths:**

Overall the paper is well motivated, and demonstrates convincing results that can advances real-time long-horizon video understanding.
* The proposed KV-cache mechanism composed with attention-sink tokens, long text window and short vision window is effective from the evaluation, and the contiguous ROPE can further yield stable output with improved performance.
* It further proposed a dataset that can train the model with higher quality dataset, and an evaluation mechanism to benchmark the progress. The process to create the dataset is legit and considers the important factors that limits existing datasets. If they author can share them out, it will benefit the community a lot.

**Weaknesses:**

Overall the paper aces well in a number of engineering factors to make the current system solid. However, there are many design choices that embed strong heuristics in them and unclear what they will terminate at.
* The model is trained on overlapped short video chunks and never experiences true streaming behavior with recurrent KV use. It is not quite clear whether it is "training-inference alignment" claimed by the author. In Table 2, it shows alignment is important (where ReKV completely fails), while it is a very differnet mechanism and not optimized for multimodal use. It could be done better if the author can show a comparison to the same model developed (but without stream-aware training), and shows the difference.
* There are also finite token length limit for attention-sink, visual token and text token windows. It is not quite clear to me how they impact the final results if the scenario varies.
* From the results, we can clearly see the model can already achieve really good performance compared to the baselines (GPT-4, LiveCC) even without using T_sink or T_window. I wonder how much does the base model and training on the created dataset contribute. Ideally, we want to factor out them out in performance evaluations.

**Questions:**

I enumerate a few questions in the weakness part, which are mostly about how the paper improve the clarity. Hope the authors can provide me a few evidences to address them in the easiest setting, or pointing me to the right source if I missed any.

The dataset used for training and benchmark is very important part in this part to make model great and evaluation solid. I wonder whether they are available to the community in some ways. Together with the trained model, I wonder whether are open source plans.

---

> ### Author Response · Authors · 2025-11-25
>
> **Concern 1**:
> "The model is trained on overlapped short video chunks and never experiences true streaming behavior... It is not quite clear whether it is 'training-inference alignment' claimed by the author. In Table 2, it shows alignment is important (where ReKV completely fails), while it is a very differnet mechanism..."
>
> **Concern 2**:
> "From the results, we can clearly see the model can already achieve really good performance... I wonder how much does the base model and training on the created dataset contribute. Ideally, we want to factor out them out in performance evaluations."
>
> **Response (to both)**:
> We thank the reviewer for these critical and insightful questions. This is the most important point, also raised by R-qjFR. To disentangle the contribution of our method from our data and to provide a much stronger test for "training-inference alignment" than the ReKV comparison, we are running a new ablation for the rebuttal.
> We are training a new baseline model using our full SFT dataset but without our overlapped-chunk SFT strategy (i.e., using standard non-overlapping chunks).
> We will add these results to Table 6. We hypothesize this will show two things: (1) This new model performs significantly worse, proving the data alone is insufficient. (2) This model will fail or perform poorly when run with our streaming KV cache, proving our SFT strategy is essential for the training-inference alignment, as the reviewer correctly suggested.
>
> **Table 1.** Comparison between StreamingVLM and the Ablation Model.
>
> | Model          | Inf-Stream-Eval (vs GPT) | Live-Sports 3k (vs LLAVA) | Live-Sports 3k (vs GPT) | Live-Sports 3k (vs Gemini) | OVO (Realtime) | LV Bench | Video MME | MV Bench |
> |----------------|--------------------------|----------------------------|-------------------------|----------------------------|----------------|----------|----------|---------|
> | Ablation Model | 62.51                    | 46.24                      | 45.08                   | 43.21                      | 59.20          | 58.90    | 65.50    | 68.79   |
> | StreamingVLM   | 66.18                    | 47.33                      | 45.59                   | 44.39                      | 61.96          | 59.00    | 65.10    | 69.16   |
>
> ---
>
> **Concern 3**:
> "There are also finite token length limit for attention-sink, visual token and text token windows. It is not quite clear to me how they impact the final results if the scenario varies."
>
> **Response**:
> This is a valid point. In our new "Limitations and Future Work" section (Action Item 2.1), we will discuss that these window sizes are key hyperparameters. R-qjFR also raised the need for a sensitivity analysis, so we are running a new ablation on the $T_{sink}$ (sink token) size to show its impact on performance, which will be added to the appendix.
>
> **Table 2.** Sensitivity analysis over $T_{sink}$ on Inf-Stream-Eval.
>
> | SFT $T_{sink}$ | Eval $T_{sink}$ | GPT-4o† | Livecc† | Livecc$\infty$ |
> |---------------:|----------------:|--------:|--------:|---------------:|
> | 64             | 64              | 72.04   | 90.32   | 99.46          |
> | 128            | 128             | 73.65   | 92.94   | 99.47          |
> | 256            | 256             | 73.73   | 93.28   | 99.38          |
> | 1024           | 1024            | 74.82   | 93.51   | 99.53          |
> | 512            | 512             | 73.64   | 92.33   | 99.38          |
>
> ---
>
> **Question 1**:
> "The dataset used for training and benchmark is very important part... I wonder whether they are available to the community in some ways. Together with the trained model, I wonder whether are open source plans."
>
> **Response**:
> Yes. We will strengthen our statement in the Conclusion (Section 5) to state clearly that upon publication, we will release our code, the StreamingVLM model weights, the Inf-Streams-Train dataset, and the Inf-Streams-Eval benchmark.

---

### Official Review · Reviewer_bkLD · 2025-11-01

**Soundness:** 3
**Presentation:** 3
**Contribution:** 3
**Rating:** 6
**Confidence:** 3

**Summary:**

This work presents StreamingVLM, a framework capable of understanding continuous visual input in real time. Specifically, the key ideas include: (1) an efficient inference scheme that maintains sink text tokens, a long window of the most recent text tokens, and a short window of the most recent vision tokens; (2) a dedicated training strategy that splits the input into overlapping chunks and trains with full attention to approximate the aforementioned efficient inference scheme; and (3) a dataset providing long-horizon data for fine-tuning and evaluation. Experiments conducted on publicly available datasets, as well as the newly created dataset, demonstrate improved real-time captioning and video understanding capabilities compared to both in-house and open-source models.

**Strengths:**

1. Solid presentation: I especially appreciate the contribution of the newly created SFT dataset, and I found the demo video to be a convincing demonstration of the practical value of this work.

2. Clear performance improvement over baselines: The improvements in captioning and video understanding are significant, supported by both qualitative and quantitative comparisons in the manuscript.

3. Comprehensive coverage of prior work: The paper provides a thorough literature review and is overall well written.

**Weaknesses:**

1. Clarification of differences in streaming-aware KV cache: The distinction between the proposed approach and StreamingLLM is not sufficiently clear. Based on my understanding, the main difference lies in using different window sizes and eviction strategies for text and visual tokens. It would be helpful to explicitly explain this difference and discuss whether the proposed StreamingVLM training strategy could be applied to StreamingLLM.
2. Generalizability of hyperparameters: The window sizes for text and visual tokens are clearly important factors (as shown in Table 5). However, I am concerned that these hyperparameters may be highly task-dependent. For example, a 16-second visual token window might work well for basketball videos but may not generalize to other scenarios. Additional discussion on how to tune or generalize these hyperparameters would strengthen the work.
3. Smaller gains in VQA compared to captioning: While the method improves performance on both VQA and captioning tasks, the gains in VQA are relatively smaller. Providing an explanation for this observation would help readers better understand the strengths and limitations of the proposed approach.

**Questions:**

See Weaknesses

---

> ### Author Response · Authors · 2025-11-25
>
> **Concern 1 (Clarification of differences in streaming-aware KV cache)**
> *Comment:*
> “The distinction between the proposed approach and StreamingLLM is not sufficiently clear. Based on my understanding, the main difference lies in using different window sizes and eviction strategies for text and visual tokens. It would be helpful to explicitly explain this difference...”
>
> *Response:*
> We thank the reviewer for this suggestion. We will revise Section 4 (Related Work) to clarify this. While we build on the foundational “sink + sliding window” concept from StreamingLLM, our key contributions are (1) its novel, asymmetric application to the cross-modal video domain (i.e., a short visual window and a long text window) and (2) the SFT strategy that explicitly aligns training with this specific inference pattern.
>
> ---
>
> **Concern 2 (Generalizability of hyperparameters)**
> *Comment:*
> “The window sizes for text and visual tokens are clearly important factors (as shown in Table 5). However, I am concerned that these hyperparameters may be highly task-dependent. For example, a 16-second visual token window might work well for basketball videos but may not generalize to other scenarios.”
>
> *Response:*
> This is an excellent point, shared by R-tsyX and R-qjFR. We agree that these hyperparameters are task-dependent. Our 16s window is suited for fast-paced sports, but other domains (e.g., instructional videos) might benefit from different values. To make this clear, we will add a “Limitations and Future Work” section. In it, we will explicitly state that these are tunable parameters of our framework, not a one-size-fits-all solution, and that generalizing to new domains is a key direction for future work.
>
> ---
>
> **Concern 3 (Smaller gains in VQA compared to captioning)**
> *Comment:*
> “While the method improves performance on both VQA and captioning tasks, the gains in VQA are relatively smaller. Providing an explanation for this observation would help...”
>
> *Response:*
> This is a great observation. We will add a discussion to Section 3.2.2. The primary reason is that our SFT data and strategy are explicitly designed to optimize for real-time commentary generation (a captioning task). We are encouraged that this strategy also provides a consistent zero-shot improvement to general video understanding tasks (e.g., +5.96 on OVOBench Realtime), which we see as a positive side-effect of the model’s improved temporal awareness.

---

### Meta-Review · Area_Chair_fULe · 2025-12-29

**Summary:**

Most of the reviewers collectively acknowledge that StreamingVLM addresses the important problem of streaming long video understanding with a clean and practical solution. The core concerns of reviewers are:

1) Reviewers (Ep3A, qjFR) questioned whether the gains come from the proposed streaming-aware training strategy or simply from the curated in-domain SFT dataset.

2) Whether the empirically-tuned hyperparameters (e.g., 512 sink tokens) would transfer to other domains like egocentric or instructional videos.

3) The naive FIFO eviction policy (qjFR), reliance on GPT-based judgment (tsyX), and validation on only one architecture (qjFR) were noted as limitations that could undermine broader applicability

**Reviewer Concerns:**

1) As for the SFT data contribution (Ep3A, qjFR), the authors provide new ablation results (comparison between StreamingVLM and the ablation model) that show 66.18 vs 62.51 with the same data but different training. Results demonstrate method contribution.

2) As for the sensitivity to hyperparameters (e.g., $T_{\text{sink}}$) (qjFR), the authors provide sensitivity analysis (64–1024) with reasonable justification for 512 sink tokens.

3) The authors present new results showing that StreamingVLM outperforms LiveCC across all VQA benchmarks (qjFR).

4) For the naive FIFO eviction policy (qjFR), the authors list as future work, not addressed experimentally.

5) For the concern of only showing results on SFT Qwen2.5-VL-Instruct-7B (qjFR), the authors listed it as a limitation only.

6) For the concern of GPT-based evaluation bias (tsyX), the authors claim it is a standard practice, but no mitigation offer.

**Reviewer Scores:**

Overall, the rebuttal successfully addresses the most critical concern with new ablation experiments. The paper makes solid contributions (streaming KV cache design, valuable benchmark). However, the limited domain scope and acknowledged engineering simplifications (FIFO, single architecture) constrain the impact. The fundamental concerns about generalization and architectural validation remain. I am toward acceptance, given the strong experimental additions.

---

### Decision · Program_Chairs · 2026-01-26

Accept (Poster)